# Immunisation with Transgenic *L. tarentolae* Expressing Gamma Glutamyl Cysteine Synthetase from Pathogenic *Leishmania* Species Protected against *L. major* and *L. donovani* Infection in a Murine Model

**DOI:** 10.3390/microorganisms11051322

**Published:** 2023-05-17

**Authors:** Derya Topuz Ata, Muattaz Hussain, Michael Jones, Jonathan Best, Martin Wiese, Katharine Christine Carter

**Affiliations:** 1Strathclyde Institute of Pharmacy and Biomedical Sciences, University of Strathclyde, Glasgow G4 0RE, UK; 2Cell Guidance Systems, Babraham Research Campus, Cambridge CB22 3AT, UK

**Keywords:** leishmaniasis, vaccine, transgenic promastigotes

## Abstract

Leishmaniasis is a protozoan disease responsible for significant morbidity and mortality. There is no recommended vaccine to protect against infection. In this study, transgenic *Leishmania tarentolae* expressing gamma glutamyl cysteine synthetase (γGCS) from three pathogenic species were produced and their ability to protect against infection determined using models of cutaneous and visceral leishmaniasis. The ability of IL-2-producing PODS^®^ to act as an adjuvant was also determined in *L. donovani* studies. Two doses of the live vaccine caused a significant reduction in *L. major* (*p* < 0.001) and *L. donovani* (*p* < 0.05) parasite burdens compared to their respective controls. In contrast, immunisation with wild type *L. tarentolae*, using the same immunisation protocol, had no effect on parasite burdens compared to infection controls. Joint treatment with IL-2-producing PODS^®^ enhanced the protective effect of the live vaccine in *L. donovani* studies. Protection was associated with a Th1 response in *L. major* and a mixed Th1/Th2 response in *L. donovani*, based on specific IgG1 and IgG2a antibody and cytokine production from in vitro proliferation assays using antigen-stimulated splenocytes. The results of this study provide further proof that γGCS should be considered a candidate vaccine for leishmaniasis.

## 1. Introduction

Leishmaniasis, a protozoan disease caused by infection with *Leishmania*, is responsible for significant morbidity and mortality. Leishmaniasis is endemic in 90 countries and up to 1.2 million new cases are reported/year, although case numbers are known to be under-reported [1]. Control of leishmaniasis relies on surveillance, vector control, limiting human exposure to vectors and treatment of active cases, as currently there is no clinical vaccine available to protect people against infection [2]. However, case numbers may increase as intended control measures may be delayed by up to 9 years due to COVID lockdowns/restrictions [3]. A number of different vaccines have been tested against leishmaniasis, including live vaccines, sub-unit protein vaccines, recombinant protein vaccines, and DNA and RNA vaccines [4,5]. Live vaccines are generally more successful as they target protective antigens to the site of the infection, express a variety of antigens, and induce the type of immune response associated with a normal infection. However, live vaccines can have safety issues as the parasites can cause active disease [6]. One way to overcome this issue is to use genetically modified parasites or to use *Leishmania* species that are not infective to humans. *L. tarentolae* is a parasite of the white-spotted wall gecko, and it is not infective to mammals. It has been used as a protein expression vector to produce heterologous intracellular or secreted proteins of human and viral origin [7]. In previous studies, we have shown that gamma glutamyl cysteine synthetase (γGCS, EC 6.3.2.2, also known as γ-L-glutamyl-L-cysteine synthetase) is a potential vaccine candidate as it is essential to the survival of *Leishmania* since it catalyses the rate-limiting step in the production of glutathione, an important anti-oxidant. Importantly, studies have shown that it is not possible to produce gene-deficient parasites that cannot express this protein, demonstrating that expression of this protein by this parasite is essential, and thus the host will be exposed to this protein *in vivo* [8]. More importantly, we have shown that targeting this protein induced protective immunity against different *Leishmania* species when used as a DNA or recombinant protein vaccine [9,10,11], but we could not induce sterile immunity. Therefore, this protein will always be expressed in the host and provide an antigen. Adjuvants can boost host immune responses and increase the efficacy of vaccines [8] but selecting the most appropriate one for a vaccine candidate is difficult. PODS^®^ is a technology for producing nanostructure protein cocrystals (200 nm–5 μm) built from the polyhedrin protein of *Bombyx mori cypovirus*. The PODS crystals (PCs) have been shown to allow slow release of their cargo protein [12]. IL-2 was selected as a PCs cargo protein as it is well known for its ability to stimulate T cell responses [13]. T cells are important in immunity to leishmaniasis [14], and their induction is associated with vaccine efficacy [4]. Human IL-2 binds to specific high affinity receptors expressed on activated mouse or human T cells, stimulating both cytotoxic and T helper responses and non-specific cells, such as natural killer cells. However, IL-2 can also have detrimental effects as it can stimulate the expression of T regulatory cells, which can suppress protective immune responses; additionally it can be directly toxic, causing vascular leakage and inducing a cytokine storm [15,16]. In this study, we assessed the ability of *L. tarentolae* promastigotes, stably expressing gamma glutamyl cysteine synthetase from *L. donovani*, *L. major* or *L. mexicana* to protect against *L. major* or *L. donovani* in a murine model. Our long term aim is to try and identify a vaccine that can be used to protect against these three parasites. We also determined whether PCs expressing IL-2 could be used as an adjuvant and increase the efficacy of our live vaccine.

## 2. Materials and Methods

### 2.1. Materials

Antibiotics were supplied by Sigma Aldrich, Irvine, UK. *Escherichia coli* strains BL21 and DH5α were supplied by Novagen, London, UK. Anti-cytokine detection antibodies were supplied by BD Biosciences, Oxford, UK. Horseradish peroxidase (HRP)-conjugated goat anti-mouse IgG1 and IgG2a was obtained from Southern Biotechnology Associates Inc, Birmingham, AL, USA, and supplied by Cambridge Bioscience, Cambridge, UK. PODS^®^-Empty (no cargo) and PODS^−^IL-2 were supplied by Cell Guidance Systems, Cambridge, UK. Anti-human and anti-mouse IL-2 were provided by BD Biosciences, New York, NY, USA. Taq™ DNA polymerase, His-Tag (D3I10), XP^®^ anti-rabbit horse radish peroxidase (HRP) antibody, anti-GFP (4B10) mouse antibody, anti-mouse IgG HRP antibody, restriction endonucleases, 1 kb DNA ladder, and prestained protein markers (7–175 kDa) were obtained from New England Biolabs, Hitchin, UK. Bio-Rad protein dye reagent was obtained from Bio-Rad Laboratories, Hertfordshire, UK. Human T Cell Nucleofector Kit was supplied from Lonza Sales Ltd., Basel, Switzerland. NucleoSpin Extract II kit was supplied from Fisher Scientific, England, UK. All other reagents were analytical grade.

### 2.2. Animals and Parasites

Male and female age-matched BALB/c (20–25 g) in-house inbred mice and male and female outbred Golden Syrian hamsters, supplied by the University of Strathclyde colony, were used in studies. The following parasite strains were used in studies: *L. donovani* (MHOM/ET/67: LV82), *L. tarentolae* (strain Parrot-TarII), *L. mexicana* (strain *Lmex*luc, derived from MNYC/BZ/M379), and *L. major* (*Lmaj*luc, derived from WHOM/IR/173). All studies had local ethical approval and United Kingdom Home Office approval (project licence PPL PF669CAE).

### 2.3. Production of Transgenic L. tarentolae Promastigotes Expressing Gamma Glutamyl Cysteine Synthetase (γGCS)

Three plasmids for DNA fragment isolation for integration into the ribosomal RNA gene locus of *Leishmania tarentolae* containing either of the gene sequences for *L. donovani*, *L. major*, or *L. mexicana* γGCS were produced (an overview of the plasmids used is shown in Appendix A). For this, pSSU-INT-lmmkk-lmcpbds-pac [17] was modified in three steps to generate pSSUMCS. The pUC-GCS-GFPB was generated by gene synthesis (Biomatik, supplied by 2BScientific Ltd., Upper Heyford, UK) to contain a new multiple cloning site (5′-CTCGAGACTAGTATCGAT**TCGCGA**CAATTGGGATCCATGCACATGCGGCCGCGTATCCACATCACCATCATCACTAGCCTAGGAGTACT**TCGCGA**AAGCTTTCTAGA-3′). The pSSU-INT-lmmkk-lmcpbds-pac plasmid and pUC-GCS-GFPB plasmid were cleaved with NdeI and XbaI, the 6687 bp and 335 bp DNA fragments were isolated and ligated to form the 7022 bp plasmid pSSU4MCS. Then, pSSU-INT-lmmkk-lmcpbds-pac and pSSU4MCS were cleaved with NdeI and XhoI, the 671 bp and 6792 bp DNA fragments isolated and ligated to generate pSSU1-6mcsHIS (7463 bp). To reduce the new MCS to 5′-CTCGAGACTAGTATCGAT**TCGCGA**AAGCTTTCTAGA-3′, this plasmid was cleaved with restriction enzyme NruI to isolate and religate the 7388 bp fragment and to form the plasmid pSSUMCS. The pUC-GCS-GFPB plasmid was cleaved with BamHI and NotI to isolate a 3191 bp DNA fragment. The *L. donovani*, *L. major*, and *L. mexicana* γGCS expression plasmids pET24aLdonGCS, pET24aLmexGCS, and pET24aLmajGCS produced inhouse were cleaved with BamHI and NotI, the 2069 bp DNA fragments isolated and ligated to the 3191 bp pUC-GCS-GFPB DNA fragment to result in the plasmids pUCLdonGCSHis, pUCLmexGCSHis, and pUCLmajGCSHis. To introduce GFP, these three plasmids and the pTHcGFPn plasmid [18] were cleaved with MfeI to isolate a 2139 bp GCS DNA fragments from the three *Leishmania* species and a 7697 bp DNA fragment, respectively. The fragments were ligated to generate the plasmids pTHGFPLdonGCSHis, pTHGFPLmexGCSHis, and pTHGFPLmajGCSHis. Finally, the different pTH plasmids were cleaved with PmeI and AvrII to generate the respective 2831 bp DNA fragments containing the GFPGCSHis sequences. These were ligated with the 7378 bp DNA fragment derived from pSSUMCS cleaved with NruI and XbaI to give the plasmids pSSUGFPLdonGCSHis, pSSUGFPLmexGCSHis, and pSSUGFPLmajGCSHis, which were used to provide the 7551 bp PacI/PmeI DNA fragments carrying the *L. donovani*, *L. major*, or *L. mexicana* GFPGCSHis gene sequence to transfect *L. tarentolae* promastigotes.

Late log-phase *L. tarentolae* promastigotes (3 × 10^7^) were sedimented by centrifugation at 5600× *g* for 2 min and resuspended in 100 μL of human T cell nucleofector solution, containing the supplement provided in the Amaxa human T cell nucleofector kit. The cell suspension was transferred to the provided cuvette and 1–5 μg of PacI/PmeI DNA fragment added. After gentle mixing, the cells were electroporated in the Amaxa Nucleofector II using program V-033 and then incubated on ice for 10 min before being transferred into 10 mL of complete HOMEM medium (HOMEM medium supplemented with 10% *v*/*v* foetal calf serum and 1% penicillin/streptomycin and 2 mM L-glutamine). The cells were incubated at 27 °C for 24 h to allow parasites to grow and then puromycin was added at 40 µM so that only transfected parasites would grow. The cultures were distributed to two 96-well plates (200 μL/well) in a 1:2 and 1:40 dilution. The plates were sealed with parafilm and incubated at 27 °C until resistant cells grew (10–14 days). The transfected parasites were cultured in the absence of antibiotic and then aliquots stored at −80 °C in 8% DMSO: 92% FCS *v*/*v* until required. The expression of GFP was used to indicate production of γGCS by transfected parasites.

Deletion of the native γGCS in *L. tarentolae* expressing γGCS from either of *L. donovani*, *L. major*, or *L. mexicana* was achieved in two steps by replacement with antibiotic resistance genes (overview shown in Appendix A). The plasmids pEX-A2-BLA-AT-ArgN and pBEHLtGCSko were generated by gene synthesis (Biomatik, ON, Canada). The resistance marker genes for hygromycin B phosphotransferase and blasticidin S-deaminase were isolated from pCR2.1Hyg carrying a PCR fragment generated as described before [19] and pEX-A2-BLA-AT-ArgN using BspHI/NheI and NcoI/NheI, respectively. The resulting 1028 bp Hyg DNA fragment and the 401 bp Bsd DNA fragment were ligated with the 3888 bp DNA fragment generated from pBEHLtGCSko cleaved with NcoI and AvrII. This generated the deletion plasmids, which were cleaved with EcoRV to liberate DNA fragments consisting of 575 bp upstream region of *L. tarentolae* γGCS, the resistance marker gene, and 351 bp of the γGCS downstream region.

### 2.4. Polymerase Chain Reaction (PCR) to Confirm Integration of Heterologous γGCS and Deletion of Native γGCS in L. tarentolae

Genomic DNA (gDNA), isolated from *L. tarentolae,* was used to assess the integration of γGCS into the ribosomal RNA gene locus. LeishSSU and GFP2.rev (Table 1) were used as primers to amplify a 1531 bp fragment to indicate correct integration of the respective construct into the ribosomal RNA gene locus. Twenty-five µL reactions containing 30 ng template DNA, 1 μL of the (100 µM) stock of each primer, 5 µL of the supplied 5× buffer that included MgCl_2_ and dNTPs and 1 unit MyTaq™ DNA polymerase (Scientific Laboratories Supplies, Newhouse, UK) were used. The PCR conditions consisted of an initial denaturation at 94 °C for 3 min, followed by 30 cycles of denaturation at 94 °C for 45 s, annealing at 54 °C for 30 s and extension at 72 °C for 80 s. A final extension was carried out at 72 °C for 90 s. Diagnostic PCR reactions under standard condition (denaturation at 94 °C for 45 s, annealing at 52 °C for 30 s and extension at 72 °C for 60 s, for 30 cycles) to confirm the deletion of native *L. tarentolae* γGCS (LtaP18.1640) by replacement with the genes for hygromycin B phosphotransferase and blasticidin S deaminase were performed using γGCS-specific primers LtUPStr-F-4 and LtWTGCS-R-5 and Blasticidin-int.rev and Hygint.rev, respectively (Table 1). Parasites, which only contained heterologous γGCS, were cultured in the absence of antibiotic and then aliquots stored at −80 °C in 8% DMSO, 92% FCS until required.

### 2.5. Promastigote Imaging Studies

Parasites were viewed using Nikon upright epifluorescence microscope E600 to detect the presence of GFP, as this would indicate the production of recombinant γGCS. Images were captured from WT and transfected parasites and analysed using the WinFluor imaging software (Version 3.8.2) to ensure that parasites did not emit autofluorescence.

### 2.6. Immunoblot Analysis

The production of heterologous γGCS by transfected *L. tarentolae* promastigotes was determined by using immunoblot analysis. Proteins were separated using SDS–PAGE then transferred to an Immobilon-P polyvinylidene difluoride (PVDF) or nitrocellulose membrane by electroblotting using a XCell SureLock^TM^ E10001 (Invitrogen^TM^, Inchinnan, UK) system at a current of 4 mA/cm^2^ of gel for 90 min. Following blotting, the membrane was incubated for one hour at 37 °C in an appropriate blocking solution (5% *w*/*v* milk powder, 0.2% Tween 20 in PBS pH 7.4) at 110 rpm in a shaking incubator. The membrane was then washed three times for 5 min in PBS pH 7.4 supplemented with 2% (*v*/*v*) Tween 20. The membrane was then incubated with the mouse anti-GFP antibody (1/2000 in blocking solution) or rabbit anti-His HRP conjugate (1/5000 in blocking solution) for one hour with gentle agitation at 37 °C. The membrane was then washed four times as before at room temperature. For the mouse anti-GFP antibody, the membrane was incubated with anti-mouse IgM HRP-linked antibody (1/2000 dilution in blocking solution) for 1 h at room temperature or 37 °C. Following three washes in PBS/Tween 20 and two in PBS pH 7.4, the membrane was developed using Novex™ ECL chemiluminescent substrate reagent kit and exposed to X-ray film (CEA medical X ray film screen, AgFa Healthcare UK Ltd, Brentford, UK) for 1 s–60 min.

### 2.7. L. major Vaccination Studies

BALB/c female mice (n = 5/treatment) were injected subcutaneously in the shaven rump on days 0 and 14 with 100 µL PBS pH 7.4 alone (control) or PBS pH 7.4 containing live wild type (WT) *L. tarentolae* parasites, *L. tarentolae* expressing different heterologous γGCS genes (1 × 10^8^/mL, *L. don* γGCS, *L. maj* γGCS, or *L. mex* γGCS), or a mixture of all three types (1 × 10^8^/mL, 1:1:1 ratio i.e., a dose consisted of 1/3rd of each type of parasite, triple vaccine, Figure 1). The triple vaccine was tested as a combination of *L. mex* γGCS and *L. don* γGCS and may protect against *L. donovani*, *L. major,* or *L. mexicana*. Two weeks after the second immunisation (day 28), mice were infected by subcutaneous injection in the footpad with 10 µL incomplete RPMI-1640 medium (RPMI 1640 supplemented with 100 μg/mL penicillin/streptomycin and 2 mM L-glutamine) containing 1 × 10^7^/mL *L. major* luciferase expressing promastigotes (*Lmaj*Luc). Parasite growth was monitored by measuring footpad thickness using a pocket thickness gauge range 9 mm (Mitutoyo Corporation, Tokyo, Japan). Spleen and popliteal lymph nodes were removed aseptically and used in lymphocyte proliferation assays. The infected footpad of each mouse was removed and disrupted in 5 mL incomplete RPMI-1640 medium using the end of a sterile 2 mL syringe through a fine gauze and the resulting supernatant, free of all debris, collected. The number of amastigotes present in the footpad supernatant was determined by viewing a sample loaded into a haemocytometer under a microscope (×400 magnification), and the number of amastigotes present/mL determined. This allowed the total number of parasites present in each footpad to be determined, i.e., in the initial 5 mL volume. 

### 2.8. L. donovani Vaccination Studies

BALB/c mice (n = 5/group) were injected subcutaneously into the loose skin over the neck on day 0 and day 21 with 0.2 mL PBS pH 7.4 alone (control), PBS containing 2 × 10^7^/mL wild type *L. tarentolae* promastigotes, 2 × 10^7^/mL transgenic *L. tarentolae* promastigotes (expressing γGCS from *L. donovani* alone or a 1:1:1 mixture of parasites expressing γGCS from three pathogenic species (triple vaccine), PODS-Empty (50 million/mouse), PODS-IL-2 alone (50 million/mouse), 2 × 10^7^/mL transgenic *L. tarentolae* promastigotes, or PODS-IL-2 (50 million/mouse, Figure 1). Three weeks later, the mice were infected by intravenous tail injection (no anesthetic) with *L. donovani* amastigotes (0.2 mL RPMI-1640 supplemented with 100 μg/mL penicillin/streptomycin and 2 mM L-glutamine, 2 × 10^8^/mL), harvested from the spleen of an infected stock hamster. Parasite burdens in all mice was assessed on day 14 following the method of Carter et al. [20]. The effect of vaccination on parasite burden was assessed by determining the reduction in spleen, liver, and bone marrow parasite burdens for each mouse compared to the mean control value. The effect of vaccination or infection on neutrophil or macrophages recruitment was determined following the method of [21]. Mice were injected intraperitoneally with luminol (150 mg/mL, saline) 3 h after immunisation in neutrophil studies and 72 h after treatment with lucigenin solution (10 mg/kg, saline) for macrophage studies. Mice were imaged 5 min after injection (medium binning, 2 min imaging) and amount of bioluminescence (total flux, photon/sec) emitted in each region of interest (ROI) was determined using the Living Image software (version 4.3.1, Caliper Life Sciences, Runcorn, UK). The same-sized region of interest was used for each mouse at each time point so that the area was the same in all studies.

### 2.9. Production of Soluble Antigen for Immunological Assays

A soluble antigen was prepared using *L. major* promastigotes, *L. donovani* promastigotes, wild type *L. tarentolae* promastigotes, or *L. tarentolae* promastigotes expressing γGCS from one of the three *Leishmania* species. Promastigotes were freeze-thawed (10^8^ parasites/mL PBS pH 7.4) five times, and the resulting suspension was centrifuged at 3000× *g*. The protein concentration in the supernatant (soluble antigen) was determined using the Bio-Rad protein dye reagent and protein standards (BSA 0.1–1 mg/mL). The concentration of the unknown sample was determined from the standard curve plotted using the protein standards by linear regression. In all cases, a correlation coefficient of >0.97 was obtained. Aliquots of the soluble antigen were stored at −20 °C until required.

### 2.10. Specific IgG1 and IgG2a Responses

Blood samples were collected over the course of the study using the method described by [9]. The mean reciprocal endpoint titres ± standard errors (SE) for each group were determined. Soluble antigen prepared from *L. major* promastigotes was used as the antigen in *L. major* studies and soluble antigen from *L. donovani* promastigotes or transgenic *L. tarentolae* promastigotes expressing γGCS from *L. donovani* for *L. donovani* studies.

### 2.11. Lymphocyte Proliferation

The method described by [9] was followed in lymphocyte proliferation studies. Spleen or lymph node cells (5 × 10^5^/well) were added to the appropriate wells of a 96-well tissue culture plate and incubated with medium alone (unstimulated controls), or soluble antigen (50 µg/mL PBS pH 7.4, *L. major*, *L. donovani* soluble antigen (50 μg/mL) prepared from *L. tarentolae* (wild type or transfected *L. tarentolae* expressing the *L. donovani* γGCS, *L. major* γGCS, or *L. mexicana* γGCS) or concanavalin A (10 μg/mL, positive control) in a final volume of 200 μL. Plates were incubated for 72 h at 37 °C in an atmosphere of 5% carbon dioxide. After 72 h, the plates were stored at −20 °C until cytokine or nitrite levels could be determined.

### 2.12. Cytokine Determination

Cytokine levels (human IL-2, murine IL-2, IL-10, IL-5, or IFN-γ) in the cell supernatants were determined using ELISA assay using anti-mouse cytokine antibodies and cytokine standards. The mean cytokine production (ng/mL) ± standard error (SE) for each treatment was determined.

### 2.13. Nitrite Determination

Nitrite levels in cell supernatants were determined using the Griess reagent [22]. The nitrite concentrations (μM) for the samples were determined from the standard curve plotted using the standards run on the same plate. The mean nitrite ± standard error (SE) for each treatment was determined.

### 2.14. Statistical Analysis

All experiments were repeated at least twice and results from one experiment is shown. Normally distributed data were analysed using a one-way analysis of variance (ANOVA) for 3 or more treatments, combined with Fisher’s LSD test *post hoc* to determine differences between treatments. Non-parametric data from *in vitro* or *in vivo* studies were analysed using Kruskal Wallis test for 3 or more treatments followed by a Dunn’s ad hoc test to determine differences between treatments using an online calculator (https://www.statskingdom.com/kruskal-wallis-calculator.html, accessed on various dates between 21 March 2022 to 6 March 2023) or the GraphPad software (version 6.0c). A *p* value of <0.05 was considered significant.

## 3. Results

### 3.1. Transgenic L. tarentolae Parasites Produced γGCS Protein

PCR studies confirmed that *L. tarentolae* had integrated the gene of γGCS from the respective *Leishmania* parasite into its ribosomal RNA gene locus (Figure 2A) and immunoblot analysis showed that parasites expressed a protein of the expected size, which was recognised by antibodies against each tag (Figure 2B). Examination of the parasites using epifluorescence microscopy showed that the recombinant fusion protein was expressed in the cytoplasm of transfected parasites (Figure 3).

### 3.2. Deletion of Wild Type γGCS from Transfected Promastigotes

The *L. tarentolae* γGCS gene sequence has considerable homology to γGCS gene sequences of the three pathogenic *Leishmania* spp. i.e., 86%, 85%, and 83% homology for *L. donovani*, *L. major,* and *L. mexicana*, respectively. This could lead to the formation of heterodimers between monomers of the different proteins and reduce protein expression [23]. Therefore, the genome of the transfected parasites was manipulated to delete both copies of the *L. tarentolae* γGCS gene. Analysis of the resulting transfectants showed that the *L. tarentolae* WT γGCS gene sequence was successfully deleted from *L. tarentolae* expressing *L. donovani* γGCS or *L. major* γGCS (Figure 4).

### 3.3. Vaccination with Transgenic L. tarentolae Parasites Protected against L. major Infection, with the Triple Vaccine Being Most Effective

Immunisation with WT *L. tarentolae* did not confer any significant protection based on assessment of parasite burdens or measurement of lesion size. Mice vaccinated with the triple vaccine offered the best protection against infection with *L. major*, demonstrated as a greater reduction in footpad thickness compared to infected control group (*p* < 0.0001, Figure 5) and the mean number of parasites present in the footpad compared to WT vaccine control (*p* < 0.05, Table 2). Similar studies showed that vaccination with the triple vaccine was also associated with a reduction in parasite burdens based on the change in footpad size associated with *L. major* infection compared to controls (*p* < 0.05), and the mean number parasites present in the infected footpad (*p* < 0.05 (Appendix A)).

Blood samples were collected over the course of the study to determine if immunity was associated with differences in specific IgG1 and IgG2a antibody titres and show whether mice had a Th1 or Th2 specific immune response. Specific IgG1 antibody levels were significantly higher in mice given the triple vaccine on day 23 compared to the other groups (*p* < 0.05); however, at the end of the study, IgG1 titres were similar to infected controls values (Figure 6A). In contrast, vaccination with the triple vaccine induced significant specific IgG2a antibodies by day 7 after the second vaccine dose compared to all the other groups (day 21, Figure 6A). By day 56, mice given the triple vaccine had even higher specific IgG2a antibody titres, and they were still significantly higher than the other groups (*p* < 0.01, Figure 6B). This difference is also reflected in the higher specific IgG2a/IgG1 ratio for the triple vaccine group at the end of the experiment (0.0, infected control, 0.66 WT vaccine, 1.6 *L.t L.don* γGCS vaccine, 1.5 *L.t L.maj* γGCS. Vaccine, 1.1 *L.t L.mex* γGCS vaccine, 11.52 triple vaccine). In a second experiment, vaccination with WT or the triple vaccine induced similar specific IgG1 levels over the course of the study, but specific IgG2a levels were significantly higher for mice given the triple vaccine compared to controls and immunised with WT parasites (*p* < 0.01, Appendix A).

Antigen- and Con A-stimulated spleen or lymph node cells from the vaccinated mice produced significantly higher levels of IFN-γ, nitrite, and IL-5 (*p* < 0.05, Figure 6) but similar levels of IL-10, compared to infected control values (IL-5 and IL-10 Appendix A). A similar significant increase in IFN-γ and nitrite (*p* < 0.01) but not IL-10 was obtained in a second study for antigen-stimulated spleen and lymph node cells from mice given the triple vaccine compared to the controls (Appendix A). Overall, the results indicate that the triple vaccine offered the highest level of protection against *L. major*, and this protection was associated with an increase in Th1-related immune responses.

### 3.4. Vaccination with the Triple Vaccine Was the Most Effective Vaccine against L. donovani

The ability of *L. tarentolae* expressing just *L. donovani* γGCS (*L.t L.don* γGCS) or a mixture of parasites expressing γGCS from all three pathogenic *Leishmania* species (triple vaccine) were then tested for their ability to protect against *L. donovani*. In addition, the effect of immunisation and challenge infection on neutrophil recruitment was assessed to determine if vaccination had any effect on innate immune responses. A higher dose (2 × 10^7^ transgenic parasites/mouse) was assessed in *L. donovani* studies as immunisation with 1 × 10^7^ transgenic parasites did not induce sterile immunity. Immunisation with *L. tarentolae* was associated with significantly higher levels of neutrophils at 3 h post-inoculation at the priming and boosting site compared to uninfected controls (*p* < 0.05), and similar levels of neutrophils were present for mice immunised with WT parasites or transgenic parasites expressing γGCS from pathogenic species (Appendix A). Appendix A, shows an example of the type of images obtained in these studies. Mice immunised with *L.t L.don* γGCS parasites (*p* < 0.05) and mice given the triple vaccine (*p* < 0.01) had significantly higher levels of neutrophils compared to infected controls (*p* < 0.05, Figure 7A) in the visceral area 3 h after infection with *L. donovani* amastigotes. Interestingly, these two groups had a significant reduction in liver parasite burdens compared to infection controls or mice immunised with WT *L. tarentolae* (*p* < 0.05, Figure 7B). Similar studies also showed that vaccination with the triple vaccine but not WT parasites was associated with a significant reduction in liver parasite burdens compared to control values in two experiments (*p* < 0.01, Appendix A). Vaccination with WT *L. tarentolae* or transgenic parasites resulted in a significant increase in specific IgG1 and IgG2a antibodies at priming and boosting (*p* < 0.05) compared to uninfected controls and infected controls on day 14 post-infection in both experiments (Figure 7C,D and Appendix A). On day 56, the IgG2a/IgG1 ratios were infected control, 1.35; WT *L. tarentolae*, 1.64; *L.t L.don* γGCS, 1.18; and triple vaccine, 0.115.

The effect of vaccination on cell-mediated immune responses was determined on day 14 post-infection by lymphocyte proliferation assays and assessing IFN-γ, IL-10, and nitrite levels in antigen- and ConA-stimulated cells. There was no clear difference in IL-10 or nitrite responses for the different groups of mice (Appendix A). ConA-stimulated cells from mice immunised with the triple vaccine produced significantly higher levels of IFN-γ compared to unstimulated cells from the same animal (*p* < 0.01, Figure 7E). In the second experiment, there was no significant difference in nitrite or IFN-γ production between control and immunised mice (Appendix A), but significantly more IL-10 was produced from unstimulated and ConA-stimulated cells from mice given the triple vaccine compared to controls (*p* < 0.01, Appendix A).

Overall these studies indicate that immunisation with the triple vaccine protected against *L. donovani* infection and protection was associated with missed Th1/Th2 responses.

### 3.5. PODS-IL-2 Can Boost Immune Responses but Did Not Increase the Protective Immunity against L. donovani Induced by Vaccination with Transgenic L. tarentolae Expressing L. donovani γGCS

Vaccination with the live vaccine did not result in sterile immunity against *L. donovani* in our studies. Therefore, the effect of joint immunisation with transgenic parasites expressing *L. donovani* γGCS (*L.t L.don* γGCS) and PODS-IL-2 was determined, as IL-2 is known to boost T cell responses. Immunisation with PODS-Empty or PODS-IL-2 did not induce a significant increase in neutrophil (3 h) or macrophage (72 h) levels at the injection site compared to uninfected controls (Appendix A). Immunisation with transgenic *L. tarentolae* elicited a significant increase in neutrophil levels at the immunisation site seen in previous studies (*p* < 0.05), with similar levels present for both groups (Appendix A). Joint treatment with PODS-IL-2 did not enhance the levels induced by immunisation with transgenic *L. tarentolae* alone (Appendix A). As expected, vaccination with *L.t L.don* γGCS parasites caused a significant reduction in liver parasite burdens (*p* < 0.001) compared to infected controls and joint immunisation with PODS-IL2 did not enhance the level of protection induced in two separate experiments (Figure 8A and Appendix A). Vaccination with *L.t L.don* γGCS transgenic parasites caused a significant increase in specific IgG2a antibody titres at priming compared to controls (*p* < 0.05, Figure 8B) whereas at boosting, specific IgG1 titres were significantly higher in mice immunised with transgenic parasites and PODS-IL-2 (*p* < 0.5, Figure 8C). On challenge, a reverse pattern occurred with mice given the vaccine alone producing higher levels of specific IgG2a compared to infected controls (*p* < 0.05) whereas mice given the vaccine and PODS-IL-2 had higher levels of specific IgG1 (*p* < 0.05, Figure 8D). On day 56, the IgG2a/IgG1 ratios were infected control, 0.02, PODS-Empty, 0.10, PODS-IL 2, 0.01, vaccine, 0.71, vaccine + PODS-IL 2, 0.25, showing that specific IgG1 was the main antibody produced. In a second experiment, immunisation with *L.t L.don* γGCS transgenic parasites boosted specific IgG1 and IgG2 responses (*p* < 0.05) and joint treatment with IL-2 did not enhance antibody responses (Appendix A). Unstimulated control cells from infected controls produced significantly lower levels of IL-10 compared to cells from mice in the other groups (*p* < 0.05, Figure 8E) and significantly higher levels of IFN-γ (*p* < 0.05, Figure 8F). In a second experiment, IL-10 levels for unstimulated were only significantly higher for cells from mice immunized with empty PODS (*p* < 0.05), and there was no clear difference in IFN-γ or nitrite production for cells from animals in different groups (Appendix A). Therefore, there was no clear immunological phenotype between vaccine efficacy and Th1/Th2 responses based on specific antibody production and cytokine production by antigen-stimulated splenocytes.

## 4. Discussion

The results of this study showed that vaccination with *L. tarentolae* expressing heterologous γGCS could protect against *L. major* and *L. donovani*. The triple vaccines, which contained parasites expressing γGCS from *L. donovani*, *L. major*, and *L. mexicana* was significantly more effective against the disease caused by *L. major* than vaccination with the same dose of parasites expressing the homologous γGCS protein of *L. major*, whereas against the disease caused by *L. donovani* both vaccines were equally effective. We can conclude that the expression of recombinant γGCS is responsible for the protection obtained as immunisation with WT *L. tarentolae* had no significant effect on parasite burdens compared to infected control. Alignment studies showed that there is 87.6% homology in the sequence of the protein for *L. donovani*, *L. major,* and *L. mexicana*, 95% identity and 97% similarity between *L. donovani* and *L. major*, 91% identity and 95% similarity between *L. donovani* and *L. mexicana,* and 89% identity and 94% similarity between *L. major* and *L. mexicana* [11]. Therefore, it is perhaps not surprising that cross-protection occurred for transgenic parasites expressing γGCS from different *Leishmania* species. The ability of the triple vaccine to induce significantly greater protection against *L. major* compared to vaccination with parasites expressing γGCS from *L. major* alone may be related to the different immune response induced by this parasite compared to *L. donovani* [24]. Indeed, the results of this study showed that protection against *L. major* was associated with a predominant Th1 response whereas protection against *L. donovani* was associated with a mixed Th1/Th2 response, based on cytokine produced by antigen-stimulated splenocytes and antibody responses. Protection against *Leishmania* infection is associated with the production of IFN γ by *Leishmania* specific T cells and the macrophage activation, resulting in the production of antimicrobial products such as superoxide and nitric oxide [25]. This may explain the higher level of protection associated with *L. major,* as antigen-stimulated spleen cells from vaccinated mice infected with *L. major* produced significantly higher amounts of nitric oxide after stimulation with specific antigen compared to spleen cells from infected controls, where nitrate levels were used as an indirect measure of nitric oxide production. In contrast, spleen cells from vaccinated mice infected with *L. donovani* produced similar amount of nitric oxide as infected control cells.

Other researchers have also shown that transgenic parasites expressing proteins from pathogenic *Leishmania* can protect against infection and their ability to be used as a recombinant protein expression system has been commercially exploited [26]. In this study, we found that immunisation with WT and transgenic *L. tarentolae* was associated with enhanced neutrophil recruitment at 72 h post-priming and boosting, but it did not enhance macrophage recruitment at 72 h post-immunisation. This may indicate that the transgenic parasites caused an initial inflammatory response but were quickly cleared. It is well known that *L. tarentolae* does not cause any clinical symptoms in mammalian hosts as it is a parasite of saurian reptiles [27]. *In vitro* studies have shown that *L. tarentolae* promastigotes are taken up by dendritic cells and macrophages and previous in vivo studies have shown that the injection of *L. tarentolae* induced neutrophil recruitment using an air pouch model system [27]. In this study, infection with WT *L. tarentolae* was associated with protection against *L. donovani* whereas immunisation with WT parasites did not induce a significant reduction in *L. donovani* parasite burdens. There are a number of differences between our studies which could explain the discrepancy in results. In the study by [26], mice were primed with 5 × 10^6^ WT promastigotes by the intraperitoneal route and challenged 6 weeks later with 5 × 10^7^ *L. donovani* promastigotes, using different strains to those used in this study. In this study, mice were primed on day 0 and boosted on day 21 by subcutaneous injection of 2 × 10^7^ wild type *L. tarentolae* promastigotes. They were then infected 21 days later with 2 × 10^7^ *L. donovani* amastigotes instead of promastigotes. We did obtain protection using transgenic parasites, but in our studies, production was not associated with the production of IFN gamma by splenocytes.

Adjuvants have been shown to significantly enhance the protection associated with vaccination, but it requires the use of an adjuvant that can boost host protective immune responses [28]. In this study, we found that joint treatment with transgenic parasites expressing *L. donovani* γGCS and PODS-IL-2 did not boost host immune responses compared to mice immunised with transgenic parasites alone. This may have been due to a number of factors. First, the dose of PODS-IL-2 may have been too low to stimulate T cells in vivo, IL-2 released from the PODS crystals could have been degraded in vivo before it was able to activate antigen-specific T cells, or the IL-2 may need to be present in the same location as antigen priming. Although PODS-IL-2 were injected together with the transgenic parasites, the released IL-2 could have separated after clearance of parasites by mononuclear cells. In the future, it would be better to treat with PODS crystals that contain γGCS from pathogenic *Leishmania* and IL-2 as this would ensure they would be released in the same local environment. Treatment of uninfected mice with recombinant IL-2 for 7 days, using a implanted osmotic pump which delivered 6.4 × 10^4^ U IL-2/day, increased the expression of IFN-γ, IL-4, and IL-10 mRNAs in the livers of euthymic and nude mice [29], indicating that exogenous IL-2 can prime cells for both Th1 and Th2 responses. Studies have shown that a single subcutaneous dose of recombinant human IL -2 has a half-life of 5.1 + 1.1 h [30] so even if the PODS crystals produced IL-2 continuously, it is likely to have diffused away from the injection site quite quickly. IL-2 treatment is often associated with toxic side effects, which have limited its therapeutic use except at low doses e.g., inducing vascular leak syndrome, hypotension, pulmonary oedema, liver cell damage, and decreased blood oxygen saturation [31].

Overall, this study confirmed previous findings showing that γGCS is a valid candidate as an antileishmanial vaccine. We have shown that *L. donovani* γGCS can protect against *L. donovani* as a recombinant protein vaccine, a DNA vaccine, and now as a recombinant protein produced by transgenic *L. tarentolae.* In addition, transgenic parasites expressing γGCS from *L. donovani*, *L. major*, and *L. mexicana* were significantly more effective against *L. major* infection. Additionally, our studies showed that IL-2 PODS did not enhance the protective effect of the *L.t L.don* γGCS vaccine against *L. donovani* in our murine model. 

## Figures and Tables

**Figure 1 microorganisms-11-01322-f001:**
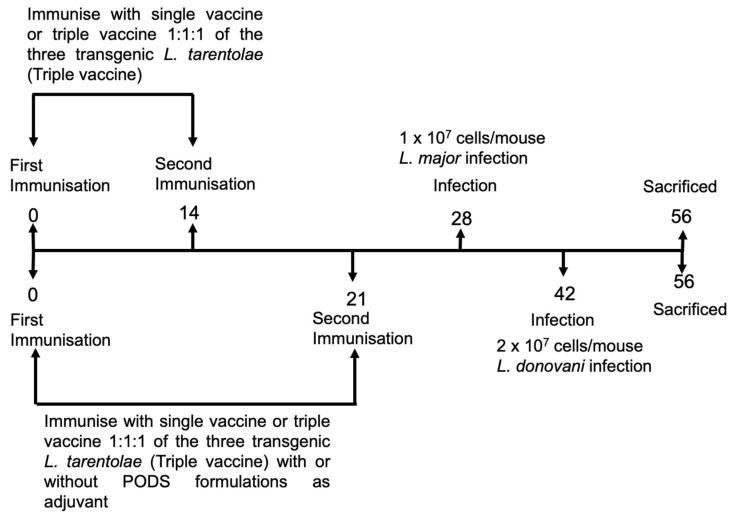
An overview of the vaccination protocol used in *L. donovani* and *L. major* experiments. Mice (n = 5/treatment) were immunised with *L. tarentolae* WT promastigotes or *L. tarentolae* promastigotes transfected with the gene sequence for *L. don* γGCS, *L. maj* γGCS, or *L. mex* γGCS alone or as a mixture of all three types of transgenic parasites (1:1:1 ratio, triple vaccine). Mice were infected on day 28 (*L. major* studies) or on day 42 (*L. donovani* studies). On day 56, mice were sacrificed, and spleens or popliteal lymph nodes were collected and used in lymphocyte proliferation assays. Blood samples were collected over the course of experiments so that antibody titres could be determined.

**Figure 2 microorganisms-11-01322-f002:**
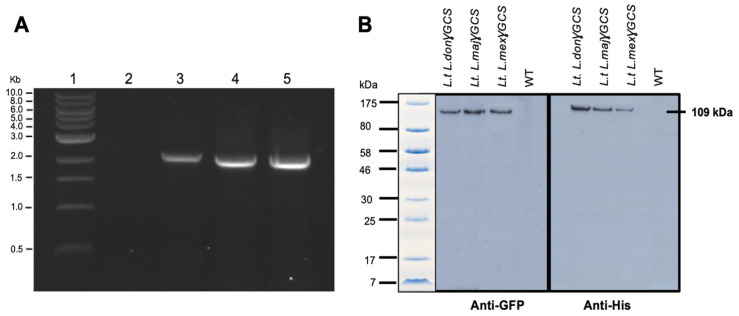
Transgenic *L. tarentolae* parasites produce heterologous double-tagged γGCS. (**A**) Agarose gel from diagnostic PCRs on genomic DNA extracted from wild type (WT) *L. tarentolae* (lane 2) or transgenic *L. tarentolae*; *L.t don* γGCS (lane 3), *L.t maj* γGCS (lane 4), or *L.t mex* γGCS (lane 5) for the detection of integration of the GFP-γGCS-His gene into the rRNA locus of the parasites. All transfected parasites showed a dominant band of (~1500 bp) whereas the WT did not produce an amplicon. (**B**) Immunoblotting of 2 × 10^7^ *L. tarentolae* (L.t.) promastigotes to detect the presence of γGCS (GCS) recombinant proteins. Proteins present in WT and transfected parasites (*L.t L. don* γGCS, *L.t L. maj* γGCS, *L.t L. mex* γGCS). A band at the expected molecular mass for a GFPGCSHis protein of 109 kDa showed with both anti-GFP and anti-His antibodies in all three transgenic *L. tarentolae* parasites. *L. tarentolae* (WT) did not express the recombinant parasite protein.

**Figure 3 microorganisms-11-01322-f003:**
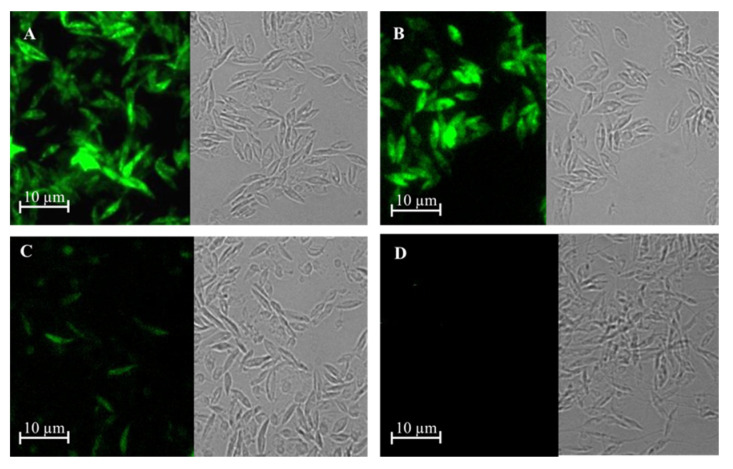
GFP expression using epifluorescence microscopy of transgenic *L. tarentolae* promastigotes. Images show expression of GFP-γGCS-His in *L. tarentolae* promastigotes transfected with *L. donovani* γGCS (**A**), *L. major* γGCS (**B**), *L. mexicana* γGCS (**C**), or wild type *L. tarentolae* (**D**) promastigotes by imaging using bright field and fluorescence microscopy.

**Figure 4 microorganisms-11-01322-f004:**
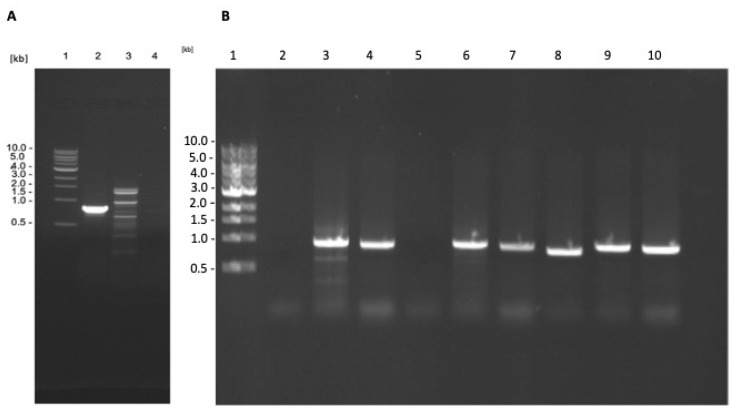
Results of the PCR assays to assess the deletion of *L. tarentolae* γGCS in parasites expressing heterologous γGCS. DNA extracted from WT *L. tarentolae* (**A**) was used in a PCR reaction using primers to detect the presence of the gene for *L. tarentolae* WT gGCS (lane 2), the hygromycin B phosphotransferase gene (lane 3), or the blasticidin-S deaminase gene (lane 4). DNA isolated from transfected parasites ((**B**), *L.t L. don γGCS*, *L.t L. maj γGCS,* or *L.t L. mex γGCS*, lane 2–4, 5–7, or 8–10, respectively) was used in a PCR reaction using primers to detect the presence of the gene of WT *L. tarentolae* γGCS (lane 2, 5, and 8), hygromycin B phosphotransferase gene (lane 3, 6, and 9) or the blasticidin S deaminase gene (lane 4, 7, and 10). The resulting PCR products were separated by using agarose gel electrophoresis and DNA bands visualised by staining with ethidium bromide. The PCR-produced bands of the anticipated size of 838 bp for WT γGCS (WTγGCS), 951 bp for hygromycin B phosphotransferase (HYG), and 917 bp for blasticidin S deaminase (BLA), molecular weight marker (lane 1).

**Figure 5 microorganisms-11-01322-f005:**
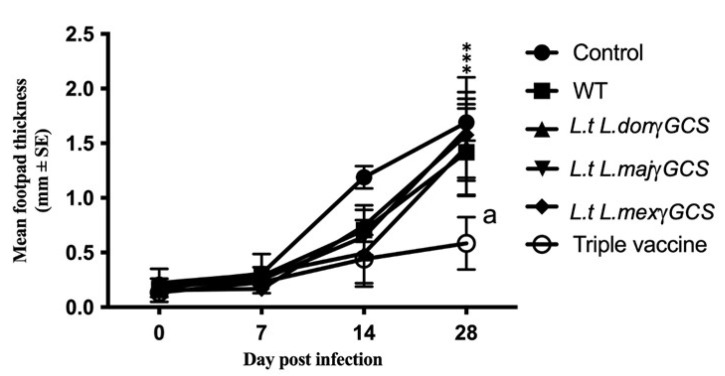
The effect of different vaccines on the parasite burdens of *L. major* infection. Mice (n = 5/treatment) were immunised on days 0 and 14 with PBS alone (control) or 1 × 10^7^ *L. tarentolae* promastigotes (WT, *L.t L.don* γGCS, *L.t L.maj* γGCS, *L.t L.mex* γGCS or a mixture of all three (triple vaccine)). On day 28, all the mice were infected with 1 × 10^7^ *L. major* promastigotes by subcutaneous injection into the footpad (*Lmaj*Luc strain) and then sacrificed on day 56. Parasite burdens were determined by assessing footpad thickness of the infected footpad relative to the uninfected footpad over the course of infection. The experiment was terminated on day 28 post-infection i.e., day 56 of the experiment. *** *p* < 0.0001 triple vaccine compared to infected control, ^a^ *p* < 0.05 triple vaccine compared to WT.

**Figure 6 microorganisms-11-01322-f006:**
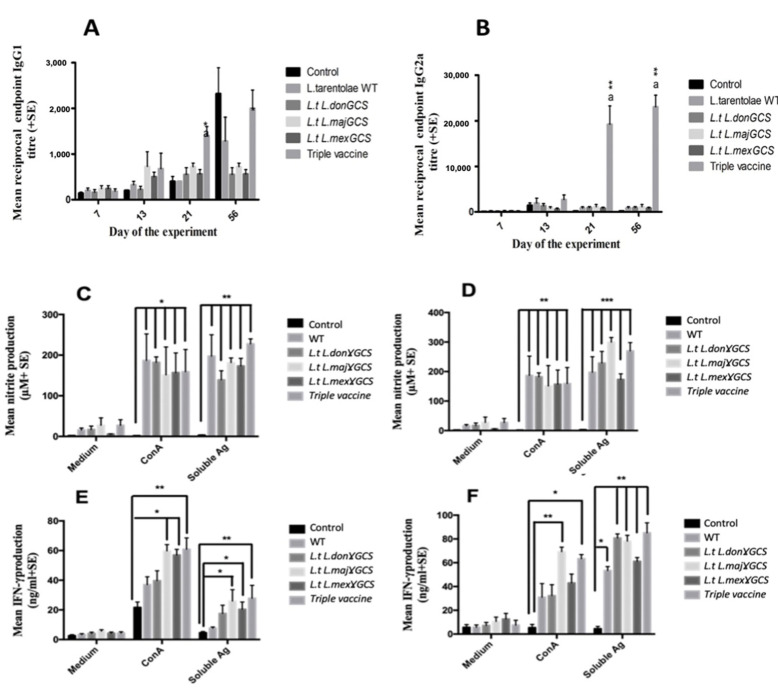
The effect of vaccination with different vaccines on the specific IgG1 (**A**), specific IgG2a (**B**) antibody titres of mice. Mice (n = 5/treatment) were immunised on days 0 and 14 with PBS alone (control) or 1 × 10^7^ *L. tarentolae* promastigotes (WT, *L.t L.don* γGCS, *L.t L.maj* γGCS, *L.t L.mex* γGCS or a mixture of all three (triple vaccine). On day 28, all the mice were infected with 1 × 10^7^
*L. major* promastigotes by subcutaneous injection into the footpad (*Lmaj*Luc strain) and then sacrificed on day 56. Mean nitrite production (**C**,**D**) and mean IFN-γ production, splenocytes (5 × 10^5^/mL (**E**)), or popliteal lymph node cells (5 × 10^5^/mL (**F**)) from the same mice were incubated with medium alone (controls), ConA (5 µg/mL) or *L. major* soluble antigen (25 µg/mL) for 72 hr. Nitrite levels in cell supernatants were determined using a Griess assay. * *p* < 0.05, ** *p* < 0.001, *** *p* < 0.0001 compared to control, ^a^ *p* < 0.01 compared to other vaccine groups.

**Figure 7 microorganisms-11-01322-f007:**
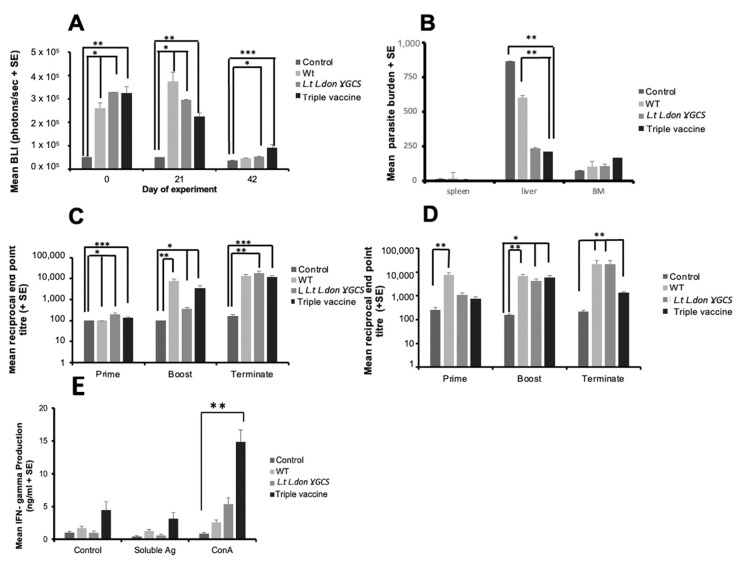
The effect of vaccination with different vaccines on immune responses and parasite burdens in mice infected with *L. donovani*. Mice (n = 5/treatment) were immunised on days 0 and 21 with PBS alone (control), 1 × 10^7^ *L. tarentolae* promastigotes (WT, *L.t L.don* γGCS or the triple vaccine). The effect of vaccination on neutrophil recruitment 3 h post-treatment, after injecting mice with luminol solution (150 mg/kg) and determining the amount of bioluminescence present at the injection site (**A**). Mice were infected on day 42 with 2 × 10^7^ *L. donovani* amastigotes and parasite burdens were determined on day 56 (**B**). Specific *L. donovani* IgG1 (**C**) and IgG2a (**D**) antibody titres were determined over the course of the study and IFN-γ production by antigen- and ConA-stimulated splenocytes (**E**) from the mice was determined on day 56. Splenocytes (5 × 10^5^/mL) from each mouse were incubated with medium alone (controls), ConA (5 µg/mL) or *L. donovani* soluble antigen (50 µg/mL) for 72 h and the amount of IFN-γ present in cell supernatants determined using ELISA. * *p* < 0.05, ** *p* < 0.001, *** *p*< 0.0001 compared to control.

**Figure 8 microorganisms-11-01322-f008:**
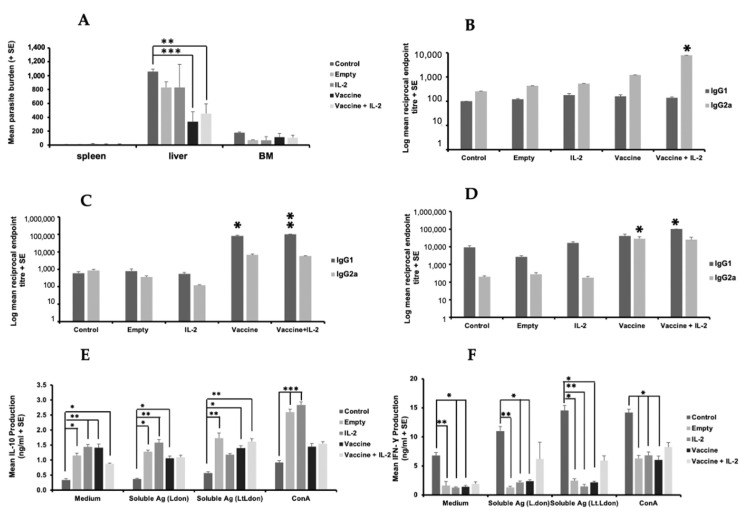
The effect of vaccination with different vaccines on parasite burdens and the immune responses of *L. donovani*-infected mice. BALB/c mice (n = 5/treatment) were immunised on days 0 and 21 with PBS alone (control), PODS-Empty (50 million/mouse), PODS-IL-2 (50 million/mouse), 1 × 10^7^ *L.t L.don* γGCS promastigotes alone (vaccine) or mixed with PODS-IL-2 (50 million/mouse, vaccine). Mice were infected on day 42 with 2 × 10^7^ *L. donovani* amastigotes and parasite burdens were determined on day 56 (**A**). The specific *L. donovani* antibody of mice on day 21 (**B**), day 42 (**C**) and day 56 (**D**) are shown. Splenocytes (5 × 10^5^/mL), prepared from mice on day 56 were incubated for 72 h with medium alone (controls), ConA (5 µg/mL), soluble antigen prepared from *L. donovani* promastigotes (50 µg/mL) or soluble antigen prepared from *L. tarentolae* expressing *L. donovani* γGCS (50 µg/mL), and IL-10 (**E**) and IFN-γ (**F**) levels in cell supernatants determined using ELISA. * *p* < 0.05, ** *p* < 0.001, *** *p* < 0.0001 compared to control. Overall, the results indicated that the live *L.t L.don* γGCS offered protection against infection and that PODS-IL-2 did not enhance the protective effect.

**Table 1 microorganisms-11-01322-t001:** Primers used in PCR studies.

**Primer Name**	**Primer Sequence**	**Use**
LeishSSU.for	Forward 5′-GATCTGGTTGATTCTGCCAGTAG-3′	Primer to determine integration of GFPGCS constructs into the ribosomal RNA gene locus of *Leishmania*
GFP2.rev	ACATGTTGGACTTGTACAGCTCGTCCAT	Primer to determine integration of GFPGCS constructs into the ribosomal RNA gene locus of *Leishmania*
LtUPStr-F-4	5′-TTCGTTGGACCTGGTTCTCA-3′	*L. tarentolae* γGCS upstream region-specific primer to determine presence of antibiotic resistance gene or wild type γGCS
LtWTGCS-R-5	Reverse 5′-CTCCTCGCCCCAAAGAAATG-3′	*L. tarentolae* γGCS specific primer to determine presence of native γGCS in LtaP18
Hygint.rev	Reverse 5′-GCAATAGGTCAGGCTCTCGC-3′	Hygromycin phosphotransferase specific primer to determine replacement of native γGCS by the resistance marker gene in LtaP18
Blasticidin-int.rev	Reverse 5’-ATCGCGACGATACAAGTCAGG-3′	Blasticidin S-deaminase specific primer to determine replacement of native γGCS by the resistance marker gene in LtaP18

**Table 2 microorganisms-11-01322-t002:** The effect of vaccination on the parasite burdens of *L. major-*infected mice. The parasite burdens of mice shown in Figure 5 were determined by direct counting of the number of parasite/mL present in infected footpads and by determining the amount of bioluminescence (BLI, photon/sec) emitted from each footpad. The mean reduction in parasite burdens ± SE compared to the infected control is shown in parentheses. The mean percentage reduction in parasite burdens ± SE compared to the infected control is shown in parentheses. * *p* < 0.05, *** *p* < 0.001 compared to infected controls, ^a^ *p* < 0.05 compared to WT group. Studies using uninfected mice showed that their footpads have a mean total flux ± SE value or 5.18 × 10^4^ ± 1.12 × 10^4^ (n = 5).

	Mean Parasite Number/mL ± SE	Mean Parasite Number/mL ± SE × 10^6^ (% Reduction ± SE Compared to Control)
*L. major* Luc parasites at infection	26 ± 0.5	N/A
Infected Control	2.98 ± 0.98	26 ± 0.58
WT vaccine	1.66 ± 0.95 (44% ± 0.18)	16 ± 0.84 (38% ± 0.11)
*L.t L.don* γGCS vaccine	0.8 ± 0.55 * (74% ± 0.02)	0.96 ± 0.14 * (96% ± 0.01)
*L.t L.maj* γGCS vaccine	1.02 ± 0.09 * (86% ± 0.01)	0.99 ± 0.2 * (94% ± 0.02)
*L.t L.mex* γGCS vaccine	1.27 ± 0.72 (72% ± 0.19)	1 ± 0.26 * (96% ± 0.01)
Triple vaccine	0.2 ± 0.17 ***^,a^ (92% ± 0.03)	0.68 ± 0.28 *^,a^ (98% ± 0.01)

## Data Availability

The data shown in this study are available on request from the corresponding author.

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
