# Peer review of "Immunisation with Transgenic L. tarentolae Expressing Gamma Glutamyl Cysteine Synthetase from Pathogenic Leishmania Species Protected against L. major and L. donovani Infection in a Murine Model"

_microorganisms, 2023, doi:10.3390/microorganisms11051322_

Round 1

Reviewer 1 Report

In the article entitled Immunisation with transgenic L. tarentolae expressing gamma glutamyl cysteine synthetase from pathogenic Leishmania species protected against L. major and L. donovani infection in a murine model, the authors study the efficacy of different vaccination strategies based on the use of modified live attenuated parasites expressing GCS from L. major, L. donovani and L. mexinaca. After a very good introduction, the article correctly details the methodology used as well as the results obtained. I have only a few minor comments that I think the authors should consider prior to publication.

- In both cases, they perform the challenge 21 days after the last immunisation. Do the authors consider that this enough time for the infection with L. tarentolae to be established, and for it to express all the proteins to generate a good and potent immune response?

- In the methodology part, the authors express in different ways the amount of promastigotes they use to both vaccinate and infect the animals (total amount vs. concentration per millilitre). Although both ways are correct, I think it would be desirable to homogenise the criteria to express it.

- Throughout the text there are several word errors with missing italics (line 223 or 281 for example).

- In accordance with the JMIR, the P in statistics is capitalised and italicised, please modify in the text.

- It is essential to improve the quality of the graphs. The work behind this article is a lot, and it is a pity that the results are not clearly visible. I advise the authors to enlarge the figures and increase the quality, even if this means splitting them up and putting more figures.

- In figure 6, instead of repeating the legend 6 times, one per figure, it is better to put a single legend common to the whole figure 6, as the colours and groups are the same, and thus the figure can be enlarged to better see the differences.

- Repeatedly, it is written "mean of..." on the ordinate axis, incorrectly, as it is not showing an axis with the mean production of ifn, for example, it is showing the production of interferon in general, and with the columns you can see the mean and SE.

- Line 398, that section seems to have two headings, keep only one.

I consider the article fit for publication, but mainly, after modifying the figures, as they do not allow to see the results properly. For the rest, I congratulate the authors for their work.

Author Response

We would like to the reviewer for taking teh time to read the manuscript and give helpful feedback. The changes  tsuggested have improved the quality and clarity of our manuscript.   We have addressed the specific points below and highlighted the changed text in the manuscript.

  1. -
    1. - In both cases, they perform the challenge 21 days after the last immunisation. Do the authors consider that this enough time for the infection with tarentolae to be established, and for it to express all the proteins to generate a good and potent immune response?

    In previous published studies we have only left 2 weeks between prime, boost and infection e.g.  Campbell et al., 2012,  Comparative assessment of a DNA and protein Leishmania donovani gamma glutamyl cysteine synthetase vaccine to cross-protect against murine cutaneous leishmaniasis caused by L. major or L. mexicana infection. Vaccine. 30:1357-63). Therefore, we think that 2 or 3 weeks is plenty of time to induce protective immunity in our model.

    1. In the methodology part, the authors express in different ways the amount of promastigotes they use to both vaccinate and infect the animals (total amount vs. concentration per millilitre). Although both ways are correct, I think it would be desirable to homogenise the criteria to express it.

    Thank you for pointing this out.  We have changed the text to show the concentration/ml used in studies (lines 215, 216, 245, 248).

    1. Throughout the text there are several word errors with missing italics (line 223 or 281 for example).

    We apologise for this mistake and we have gone through the manuscript and tried to correct all the errors.

    1. In accordance with the JMIR, the P in statistics is capitalised and italicised, please modify in the text.

    We apologise for this oversight.  I am afraid we have used this format in previous published studies,  We have changed all the P’s to a capital and initalised it throughout the manuscript.

    1. It is essential to improve the quality of the graphs. The work behind this article is a lot, and it is a pity that the results are not clearly visible. I advise the authors to enlarge the figures and increase the quality, even if this means splitting them up and putting more figures.

    We apologise for the poor quality of the figures we have tried to address this problem. We have attached the figures as ‘jpeg’ files so that the editor can let us know if they are of suitable quality for publication.

    1. In figure 6, instead of repeating the legend 6 times, one per figure, it is better to put a single legend common to the whole figure 6, as the colours and groups are the same, and thus the figure can be enlarged to better see the differences.

    We have changed the figure legend and the we have made sure the colours now match

    Repeatedly, it is written "mean of..." on the ordinate axis, incorrectly, as it is not showing an axis with the mean production of ifn, for example, it is showing the production of interferon in general, and with the columns you can see the mean and SE.

    We have used this format in previously published papers as the data could be individual readings or median so we wanted to be very clear in what we are expressing on this figure. We think this is a clearer way of expression out data.  We will take guidance from the editor on whether we have to change this text on all the figures.

    1. Line 398, that section seems to have two headings, keep only one.

    We apologise for this error and have amended the text (line 445-446)

Reviewer 2 Report

In the manuscript submitted to Microorganisms (id nº microorganisms-2296211), entitled “Immunisation with transgenic L. tarentolae expressing gamma glutamyl cysteine synthetase from pathogenic Leishmania species protected against L. major and L. donovani infection in a murine model”, by Popuz Ata et al. the authors propose a new live vaccine approach based on L. tarentolae parasites expressing heterologous gamma glutamyl cysteine. First of all, it is noticeable that the article was written by non-English native speakers. In some instances the message is not clear; therefore, I suggest you to ask someone proficient in English to read the manuscript and correct the English. Additionally, in my opinion, the methodology/rationale is not clear.  I leave below a few specific comments for you to have a change to improve your manuscript (should you agree with them). I also ask some clarifications which in my view are needed to make a decision on this manuscript.

Major points:

1-     One of the main questions I have, which is not clear in the manuscript text is in how many independent experiments was this paper (and each individual figure) based on? Additionally, how many animals were used per group?

2-     I have also many doubts with respect to the experimental scheme:

a.      For the triple vaccine you used 1 + 1 +1 X10^8 parasites, or 0.33 + 0.33 + 0.33 x10^7 parasites?

b.      Why are the schemes different with respect to L. major and L. donovani infections? Please explain the rationale behind each experiment in the manuscript. Particularly, you immunized with different parasite amounts, at different intervals, and also why did you wait so few days in the context of visceral disease (15 days is not enough to assess the consequence of effective memory responses).

3-     Please provide an explanation for the inclusion of L. mexicana-expressing ggcs parasites, and for the difference in terms of GFP expression (Figure 3), comparing with the other parasite lines generated.

4-     Please include all “data not shown” as supplementary figures.

5-     In Figure 4 an important control is missing: WT L. tarentolae parasites. Please show the respective data.

6-     With respect to bioluminescence measurements, in my opinion, there are several inaccuracies:

a.      I would like to see representative images as supplementary files;

b.      Accurate bioluminescence values should be given as average radiance and not total flux;

c.      It is not clear to me how did you obtain the values in table 2, particularly the mean parasite number. Ideally, I would like to see data based on limiting dilution, the gold standard method.

7-     With respect to Figure 6, I would like to see the data normalized over the control. Additionally, in my view, ConA is a positive control. Therefore, to base any conclusions (vaccine-induced phenotype as you mention), based on the ConA condition is inaccurate.

8-     With respect to the visceral model, I don’t know if you can talk in protection. You barely have parasites in the spleen (which is weird since the spleen is the major target organ where parasites will not be controlled), and in the liver 15 days is too early to account for an effective cellular response singe granulomas start maturing from 15 days to one month post-infection. Overall, these data (including the very low number of parasites per organ makes me conclude this model is inadequate to answer to your question.

9-     I think the quantification of “innate effectors” is a confounding factor. I would not mention neutrophils or macrophages without a clear explanation for the phenotype.

10-   I would not include the data in the adjuvant since it does not add anything to the rest of the story.

Do note that the IFNg phenotype is quite different comparing to the one determined for the triple vaccine in the previous experiment. Do you have any explanation for this?

Minor points:

1-     The use of English language should be polished.

a.      E.g. Page 1, line 11: Should read “…disease responsible for significant…”;

b.      E.g. Page 1, line 14: Should read “…their ability to protect against infection…”;

c.      E.g Page 1, line 36: Should read “…including live vaccines…”;

d.      E.g Page 2, line 52: Should read “have been shown to allow the slow release…”;

e.      And many other examples throughout the paper.

2-     Sub-section 2.3 of the Methods section is extremely confuse. I advise you to create a schematic of all the constructs used/generated, including the map of primers, and simplify the text.

Author Response

We would like to thank the reviewer for taking the time to review this manuscript and giving helpful feedback.   The changes they have suggested have improved the quality and clarity of our manuscript.   We have addressed the specific points below and highlighted the changed text in the manuscript.

Major points:

  • One of the main questions I have, which is not clear in the manuscript text is in how many independent experiments was this paper (and each individual figure) based on? Additionally, how many animals were used per group?

Every experiment was repeated at least twice but sometimes experiments were repeated 3 times. We apologise for this oversight.   We will add the number of replicates to the figure legend.  We gave the number animals of animals/treatment is given in the Materials and methods section (line 212 ‘BALB/c female mice (n = 5/treatment)’, line 240 ‘BALB/c mice (n = 5/group)’), and the number of animals/group is also stated in the figure legends (e.g. Fig 1 legend, line 262). So, if an experiment had 6 groups in it then there were 30 mice in the experiment.

  1. I have also many doubts with respect to the experimental scheme:
  2. For the triple vaccine you used 1 + 1 +1 X10^8 parasites, or 0.33 + 0.33 + 0.33 x10^7 parasites?

In the triple vaccine we thought it was important to administer the same dose of transgenic parasites rather than using three times the dose as this would not be a good ‘head to head’ comparison for using juts one type of transgenic parasite. Each mouse therefore was given the appropriate number of transgenic parasites made up of 1/3rd of each type of transgenic parasites.  We considered this the best ‘head to head’ comparison for determining whether using one type of parasite was the better option. We have amended the text to try and make this clearer (line 216-218).

  1. Why are the schemes different with respect to major and L. donovani infections? Please explain the rationale behind each experiment in the manuscript. Particularly, you immunized with different parasite amounts, at different intervals, and also why did you wait so few days in the context of visceral disease (15 days is not enough to assess the consequence of effective memory responses).

We have added a sentence in the results to explain the difference in parasite dose ‘A higher dose (2 x 107 transgenic parasites/mouse) was assessed in  L. donovani studies as immunisation with 1 x 107 transgenic parasites did not induce sterile immunity’ (line 451-453).

In previous published studies we have only left 2 weeks between prime, boost and infection and seen good protection so we initially used this protocol e.g.  Campbell et al., 2012,  Comparative assessment of a DNA and protein Leishmania donovani gamma glutamyl cysteine synthetase vaccine to cross-protect against murine cutaneous leishmaniasis caused by L. major or L. mexicana infection. Vaccine. 30:1357-63).  In this study we had good protection in the L. major model so using two weeks between doses gave sufficient to induce a protective response.  Obviously in other species the best protocol would have to be determined.

 The reason we did not use 2 weeks during vaccine studies with L. donovani was primarily because of COVID restrictions on access to the animal unit during experiments.  In our first experiment we had to have 3 week between priming, boosting and infection.  Therefore, in the repeat experiments we used the same protocol. I had considered adding this to the manuscript to justify the difference.

Please provide an explanation for the inclusion of L. mexicana-expressing ggcs parasites, and for the difference in terms of GFP expression (Figure 3), comparing with the other parasite lines generated.

One aim of vaccine studies is to try and produce a vaccine that can be used for these three species.  We have added a statement to justify the use of transgenic parasites expressing gGCS (lines 70-71 and 217-218).  Fig 3 shows GFP expression for a representative of each of the three types of parasites.  It is intended to show that the parasites successfully expressed the fusion GFP-gGCS, it is not intended as a study of the amount of protein expressed by each type of parasite.  Protein expression can vary between cells types and I am not sure how you ensure that each parasites expresses the same amount of fusion protein on a particular day.  In our study we used parasites from the culture to produce the single vaccine as the triple vaccine to try and minimise difference in fusion protein expression.  It would take GMP studies to identify a product for clinical studies.

  1. Please include all “data not shown” as supplementary figures.

We apologise for not including this data and it has now been added as supplementary data.  

  1. In Figure 4 an important control is missing: WT L. tarentolae parasites. Please show the respective data.

We apologise for missing the WT PCR data in this Fig. and this has now been added to Fig. 4.

  1. With respect to bioluminescence measurements, in my opinion, there are several inaccuracies:

  1. I would like to see representative images as supplementary files

We have published data without showing images in previous publications and that is why we did not provide images of the footpads of mice infected with luciferase-expressing parasites in these studies e.g. Alsaadi et al. 2012 (Journal of Controlled Release, 160: 685-691).  We can add representative images if required but I do not think this adds anything to this study as IVIS imaging is a commonly used technology now.

  1. Accurate bioluminescence values should be given as average radiance and not total flux

In the L. major studies,  we infected mice with luciferase-expressing parasites and this allowed us to monitor the growth of parasites in the same mouse over time (e.g. e.g. Alsaadi et al. 2012 (Journal of Controlled Release, 160: 685-691).  We are measuring bioluminescence not fluorescence in these studies and we have used the same total flux (photons/sec) as the unit, just like our previous studies.

  1. It is not clear to me how did you obtain the values in table 2, particularly the mean parasite number. Ideally, I would like to see data based on limiting dilution, the gold standard method.

Many studies would use changes in footpad size as a gold standard for monitoring the growth of L. major in mice, as it is a direct measurement over time in an experiment.  At the end of the experiment we removed the infected foot of a L. major infected mouse, mashed it up in medium to release the parasites present (amastigotes) and then counted the number of parasites present/ml using a microscope and haemocytomer (lines 229-237).  We knew the volume used to produce the homogenate (5 ml) and so we could then determine the total number of amastigotes present in the infected footpad.  We have rewritten this method to try and make this clearer.   This method is a direct count and does not depend on growing parasites in culture medium after dilution.  There was a clear difference in the number of parasites present in vaccinated and control mice based on IVIS studies over time, footpad size over the course of the study and number of parasites in the infected footpad. 

  1. With respect to Figure 6, I would like to see the data normalized over the control. Additionally, in my view, ConA is a positive control. Therefore, to base any conclusions (vaccine-induced phenotype as you mention), based on the ConA condition is inaccurate.

We have always shown the actual data we have obtained from studies so that researchers can see the data, rather than manipulated e.g. Carter et al.1999, Clin Diagn Lab Immunol. 6: 61-5.  I have always believed that as well as being a positive control, ConA shows the overall response of the cellular response and is influenced by factors e.g. infection.  I believe that in our study there is a clearer phenotype for L. major than there is for L. donovani  and this does not surprise me as L. major is often used a model for clear Th1/Th2 responses.

8.With respect to the visceral model, I don’t know if you can talk in protection. You barely have parasites in the spleen (which is weird since the spleen is the major target organ where parasites will not be controlled), and in the liver 15 days is too early to account for an effective cellular response singe granulomas start maturing from 15 days to one month post-infection. Overall, these data (including the very low number of parasites per organ makes me conclude this model is inadequate to answer to your question.

       I completely agree that that the conditions you use in your model can affect your results.  Some studies on L. donovani used much longer term experiments. month.  In a previous vaccine study we assessed parasite burdens on both day 14 and day 39 post-infection (Carter et al., 2007, Vaccine. 25:4502-9).   In that study we found vaccination gave protection at both time points so we only used day 14 in this study as this study is a less severe disease model for the mice, and animal welfare is an important consideration in our studies.  Other researchers would argue that only the hamster model is relevant in L. donovani studies.

9.I think the quantification of “innate effectors” is a confounding factor. I would not mention neutrophils or macrophages without a clear explanation for the phenotype.

We measured the neutrophil and macrophage influx to determine if the vaccines had an effect.  We agree that we did not find a clear phenotype but we think other researchers may find this interesting.

10.-   I would not include the data in the adjuvant since it does not add anything to the rest of the story.

      We understand this viewpoint but we think other researchers may find this data interesting.

  1. Do note that the IFNg phenotype is quite different comparing to the one determined for the triple vaccine in the previous experiment. Do you have any explanation for this?

We did notice this but unfortunately we have no explanation.  In L. donovani experiments we did not get the clear phenotype we had in L. major experiments.

Minor points:

  • The use of English language should be polished.

We apologise for these mistakes.

Page 1, line 11: Should read “…disease responsible for significant…”;

We have rewritten this sentence

Page 1, line 14: Should read “…their ability to protect against infection…”;

We have rewritten this sentence

Page 1, line 36: Should read “…including live vaccines…”;

We have rewritten this sentence

Page 2, line 52: Should read “have been shown to allow the slow release…”;

We have rewritten this sentence

  • Sub-section 2.3 of the Methods section is extremely confuse. I advise you to create a schematic of all the constructs used/generated, including the map of primers, and simplify the text.

We have modifed the text (lines 104-134). and provided schematic diagrams in the supplementary data (Figs S1-S4)

Reviewer 3 Report

This is very interesting work concerning the prove of concept that transgenic parasites expressing gamma glutamyl cysteine synthetase from L. donovani, L. major, and L. mexicana were significantly more effective against L. major infection.

Author Response

We would like to thank the reviewer for taking the time to review this manuscript and the encouraging comments. 

Reviewer 4 Report

This is an interesting and nicely written manuscript that I enjoyed reading; the use of wide-type L. tarentolae as the control is important.  I do have a few suggestions as indicated below:

1.  add persons 1-3 sentences (perhaps around line 60) in which you indicate the important role for gamma glutamyl cysteine synthetase, its regulation is Leishmania, and thus why you think it is a good 'target'; also give the EC number for the enzyme.

line 153 write out: Twenty five rather than use 25 

line 168: remove 'with'

line 355: replace 'if' with 'is'

Figure 5:  please make this larger so that we can see the data points more readily

line 485: add 'heterologous following 'expressing'

Line 546: this sentence is not clearly written.

Author Response

We would like to thank the reviewer for taking the time to review this manuscript and the helpful feedback.   The changes they have suggested have improved the quality and clarity of our manuscript.   We have addressed the specific points below and highlighted the changed text in the manuscript.

  1. Add persons 1-3 sentences (perhaps around line 60) in which you indicate the important role for gamma glutamyl cysteine synthetase, its regulation is Leishmania, and thus why you think it is a good 'target'; also give the EC number for the enzyme.

We have added additional text in the introduction to explain why we think this protein  is a good target in vaccine studies and the EC number for this enzyme (lines 45-53).

  1. line 153 write out: Twenty five rather than use 25 

We have changed the text to address this issue (line 168)

  1. line 168: remove 'with'

We have changed the text to address this issue (line 184)

  1. line 355: replace 'if' with 'is'

We have changed the text to address this issue (line 398).

  1. Figure 5:  please make this larger so that we can see the data points more readily

Thank you for your comments which match the other referee.  We have tried to improve the quality of our Figs.

  1. line 485: add 'heterologous following 'expressing'

We have changed the text to address this issue (line 544).

  1. Line 546: this sentence is not clearly written.

We apologise for the lack of clarity in the writing.  We have amended  the text to try and make things clearer (lines 598-600).

Round 2

Reviewer 2 Report

I read your rebuttal carefully, and, although some of my concerns were addressed, I am still not satisfied. Please see details below:

1 - If at least 2 independent experiments were performed, the figures should reflect that. Don't show one representative experiment; show the combination of all data from different experiments. Additionally, if you could represent the data as dot plots, with one symbol per animal, that would be ideal.

2 - With regards to the bioluminescence measurements, the analysis is still not adequate:

 2.1 - I would like to see representative images to try to understand how you have defined the regions of interest.

2.2 - The definition of the regions of interest is very important, and what allows you to use the most accurate units with respect to bioluminescence - average radiance (not a fluorescence unit) that takes into account the fotons per second per area and is comparable between different time-points even if the binning, exposure time... are different. For accuracy, you need to update the bioluminescence data.

2.3 - I find it pretty odd that the values of mean BLI/parasite are different between groups. You are detecting the same parasite strain (L. major). Therefore, this value should be similar among all conditions. This makes me wonder that, either the bioluminescence analysis needs refinement, or the parasite counting, as you have done it, was not a good estimation. I can only imagine it must be quite difficult (if not impossible) to count amastigotes in the context of a footpad cell suspension using a Newbauer chamber.

Please take my comments as constructive feedback.

Author Response

We apologise for not addressing all your queries and we have amended the manuscript as detailed below:

1 - If at least 2 independent experiments were performed, the figures should reflect that. Don't show one representative experiment; show the combination of all data from different experiments. Additionally, if you could represent the data as dot plots, with one symbol per animal, that would be ideal.

In many published studies authors show data from one experiment as a representative of the studies completed. If you would like us to show individual data points for each animal then some of the figures will be very busy.  We show the mean and the standard error to show the variability in the data and I have never pooled data from different experiments. We carried out statistical analysis to show significant difference between groups.  We are simply trying to show the data in the easiest way for the reader and it is how we have published previous results in a number of journals. We have attached how Fig 5 would appear if we showed individual mouse data -  we do not think this is a clearer way of presenting the data.  Therefore, we would like to leave the all data in its present form.

2 - With regards to the bioluminescence measurements, the analysis is still not adequate:

 2.1 - I would like to see representative images to try to understand how you have defined the regions of interest and  2.2 - The definition of the regions of interest is very important, and what allows you to use the most accurate units with respect to bioluminescence - average radiance (not a fluorescence unit) that takes into account the fotons per second per area and is comparable between different time-points even if the binning, exposure time... are different. For accuracy, you need to update the bioluminescence data.

We have now shown two images to give an indication of the region of interest used  studies in the supplementary data (Fig. S5 and S6) and we have published this type of data before (Alsaadi et al., 2012,  J Control Release. 160:685-91).  We have also added more details of the settings and region of interest we use in studies to the relevant methods section.  We used the same sized region of interest, and an exposure time of 2 minutes for our murine studies over the course of studies for each mouse so we copy over the ROI on different days so that the area used in the same one each mouse.  In in vitro studies we use the grid setting provided by the IVIS software and line this up with the 96 well plate so that the region of interest is the same. Therefore, I believe we can use the photon/sec reading rather than average radiance as we use the same area for the region of interest – thus the area used is a constant in our studies.

2.3 - I find it pretty odd that the values of mean BLI/parasite are different between groups. You are detecting the same parasite strain (L. major). Therefore, this value should be similar among all conditions. This makes me wonder that, either the bioluminescence analysis needs refinement, or the parasite counting, as you have done it, was not a good estimation. I can only imagine it must be quite difficult (if not impossible) to count amastigotes in the context of a footpad cell suspension using a Newbauer chamber.

 I find it a bit strange that it would be acceptable to use this starting material to set up in vitro cultures to assess parasite burdens in animals using a limiting dilution method but it cannot be used for direct parasite counting.  My understanding is that in the limiting dilution assays, replicates are set up for each mouse, and it is assumed that the first concentration for duplicates contains the same number of parasites for each mouse.   In our study we found that the amount of bioluminescence/parasite varied between 0.12-1.32 photon/sec for amastigotes whereas the promastigotes used at infection had a value of 2.2 photons/sec. We believe that our results indicate that amastigotes produce less bioluminescence than promastigotes.  This was not unexpected as the amastigote stage is smaller than the promastigote stage. The variability in bioluminescence between the treatments but this is not an unreasonable result as one would anticipate that protein expression would vary within a parasite population. 
